# Learning with SGD and Random Features

**Luigi Carratino**[*]
University of Genoa,
Genoa, Italy

**Alessandro Rudi**
INRIA – Sierra Project-team,
École Normale Supérieure, Paris

**Lorenzo Rosasco**
University of Genoa,
LCSL – IIT & MIT

## Abstract

Sketching and stochastic gradient methods are arguably the most common techniques to derive efficient large scale learning algorithms. In this paper, we investigate their application in the context of nonparametric statistical learning. More precisely, we study the estimator defined by stochastic gradient with mini batches and random features. The latter can be seen as form of nonlinear sketching and used to define approximate kernel methods. The considered estimator is not explicitly penalized/constrained and regularization is implicit. Indeed, our study highlights how different parameters, such as number of features, iterations, step-size and mini-batch size control the learning properties of the solutions. We do this by deriving optimal finite sample bounds, under standard assumptions. The obtained results are corroborated and illustrated by numerical experiments.

## 1 Introduction

The interplay between statistical and computational performances is key for modern machine learning algorithms [1]. On the one hand, the ultimate goal is to achieve the best possible prediction error. On the other hand, budgeted computational resources need be factored in, while designing algorithms. Indeed, time and especially memory requirements are unavoidable constraints, especially in large-scale problems.

In this view, stochastic gradient methods [2] and sketching techniques [3] have emerged as fundamental algorithmic tools. Stochastic gradient methods allow to process data points individually, or in small batches, keeping good convergence rates, while reducing computational complexity [4]. Sketching techniques allow to reduce data-dimensionality, hence memory requirements, by random projections [3]. Combining the benefits of both methods is tempting and indeed it has attracted much attention, see [5] and references therein.

In this paper, we investigate these ideas for nonparametric learning. Within a least squares framework, we consider an estimator defined by mini-batched stochastic gradients and random features [6]. The latter are typically defined by nonlinear sketching: random projections followed by a component-wise nonlinearity [3]. They can be seen as shallow networks with random weights [7], but also as approximate kernel methods [8]. Indeed, random features provide a standard approach to overcome the memory bottleneck that prevents large-scale applications of kernel methods. The theory of reproducing kernel Hilbert spaces [9] provides a rigorous mathematical framework to study the properties of stochastic gradient method with random features. The approach we consider is not based on penalizations or explicit constraints; regularization is implicit and controlled by different parameters. In particular, our analysis shows how the number of random features, iterations, step-size and mini-batch size control the stability and learning properties of the solution. By deriving finite sample bounds, we investigate how optimal learning rates can be achieved with different parameter choices. In particular, we show that similarly to ridge regression [10], a number of random features proportional to the square root of the number of samples suffices for $O(1/\sqrt{n})$ error bounds.

---

[*]Email: luigi.carratino@dibris.unige.it

The rest of the paper is organized as follows. We introduce problem, background and the proposed algorithm in section 2. We present our main results in section 3 and illustrate numerical experiments in section 4.

**Notation**: For any $T \in \mathbb{N}_+$ we denote by $[T]$ the set $\{1, \ldots, T\}$, for any $a, b \in \mathbb{R}$ we denote by $a \vee b$ the maximum between $a$ and $b$ and with $\wedge$ the minimum. For any linear operator $A$ and $\lambda \in \mathbb{R}$ we denote by $A_\lambda$ the operator $(A + \lambda I)$ if not explicitly defined differently. When $A$ is a bounded self-adjoint linear operator on a Hilbert space, we denote by $\lambda_{\max}(A)$ the biggest eigenvalue of $A$.

## 2  Learning with Stochastic Gradients and Random Features

In this section, we present the setting and discuss the learning algorithm we consider.
The problem we study is supervised statistical learning with squared loss [11]. Given a probability space $X \times \mathbb{R}$ with distribution $\rho$ the problem is to solve

$$\min_f \mathcal{E}(f), \qquad \mathcal{E}(f) = \int (f(x) - y)^2 d\rho(x, y), \tag{1}$$

given only a training set of pairs $(x_i, y_i)_i^n \in (X \times \mathbb{R})^n$, $n \in \mathbb{N}$, sampled independently according to $\rho$. Here the minimum is intended over all functions for which the above integral is well defined and $\rho$ is assumed fixed but known only through the samples.
In practice, the search for a solution needs to be restricted to a suitable space of hypothesis to allow efficient computations and reliable estimation [12]. In this paper, we consider functions of the form

$$f(x) = \langle w, \phi_M(x) \rangle, \quad \forall x \in X, \tag{2}$$

where $w \in \mathbb{R}^M$ and $\phi_M : X \to \mathbb{R}^M$, $M \in \mathbb{N}$, denotes a family of finite dimensional feature maps, see below. Further, we consider a mini-batch stochastic gradient method to estimate the coefficients from data,

$$\widehat{w}_1 = 0; \qquad \widehat{w}_{t+1} = \widehat{w}_t - \gamma_t \frac{1}{b} \sum_{i=b(t-1)+1}^{bt} \left( \langle \widehat{w}_t, \phi_M(x_{j_i}) \rangle - y_{j_i} \right) \phi_M(x_{j_i}), \qquad t = 1, \ldots, T. \tag{3}$$

Here $T \in \mathbb{N}$ is the number of iterations and $J = \{j_1, \ldots, j_{bT}\}$ denotes the strategy to select training set points. In particular, in this work we assume the points to be drawn uniformly at random with replacement. Note that given this sampling strategy, one *pass* over the data is reached on average after $\lceil \frac{n}{b} \rceil$ iterations. Our analysis allows to consider multiple as well as single passes. For $b = 1$ the above algorithm reduces to a simple stochastic gradient iteration. For $b > 1$ it is a mini-batch version, where $b$ points are used in each iteration to compute a gradient estimate. The parameter $\gamma_t$ is the step-size.
The algorithm requires specifying different parameters. In the following, we study how their choice is related and can be performed to achieve optimal learning bounds. Before doing this, we further discuss the class of feature maps we consider.

### 2.1  From Sketching to Random Features, from Shallow Nets to Kernels

In this paper, we are interested in a particular class of feature maps, namely random features [6]. A simple example is obtained by sketching the input data. Assume $X \subseteq \mathbb{R}^D$ and

$$\phi_M(x) = (\langle x, s_1 \rangle, \ldots, \langle x, s_M \rangle),$$

where $s_1, \ldots, s_M \in \mathbb{R}^D$ is a set of identical and independent random vectors [13]. More generally, we can consider features obtained by nonlinear sketching

$$\phi_M(x) = (\sigma(\langle x, s_1 \rangle), \ldots, \sigma(\langle x, s_M \rangle)), \tag{4}$$

where $\sigma : \mathbb{R} \to \mathbb{R}$ is a nonlinear function, for example $\sigma(a) = \cos(a)$ [6], $\sigma(a) = |a|_+ = \max(a, 0)$, $a \in \mathbb{R}$ [7]. If we write the corresponding function (2) explicitly, we get

$$f(x) = \sum_{j=1}^M w^j \sigma(\langle s_j, x \rangle), \quad \forall x \in X. \tag{5}$$

that is as shallow neural nets with random weights [7] (offsets can be added easily).
For many examples of random features the inner product,

$$\langle \phi_M(x), \phi_M(x') \rangle = \sum_{j=1}^{M} \sigma(\langle x, s_j \rangle) \sigma(\langle x', s_j \rangle), \qquad (6)$$

can be shown to converge to a corresponding positive definite kernel $k$ as $M$ tends to infinity [6, 14]. We now show some examples of kernels determined by specific choices of random features.

**Example 1** (Random features and kernel). *Let $\sigma(a) = \cos(a)$ and consider $(\langle x, s \rangle + b)$ in place of the inner product $\langle x, s \rangle$, with $s$ drawn from a standard Gaussian distribution with variance $\sigma^2$, and $b$ uniformly from $[0, 2\pi]$. These are the so called Fourier random features and recover the Gaussian kernel $k(x, x') = e^{-\|x-x'\|^2/2\sigma^2}$ [6] as $M$ increases. If instead $\sigma(a) = a$, and the $s$ is sampled according to a standard Gaussian the linear kernel $k(x, x') = \sigma^2 \langle x, x' \rangle$ is recovered in the limit. [15].*

These last observations allow to establish a connection with kernel methods [10] and the theory of reproducing kernel Hilbert spaces [9]. Recall that a reproducing kernel Hilbert space $\mathcal{H}$ is a Hilbert space of functions for which there is a symmetric positive definite function[2] $k : X \times X \to \mathbb{R}$ called reproducing kernel, such that $k(x, \cdot) \in \mathcal{H}$ and $\langle f, k(x, \cdot) \rangle = f(x)$ for all $f \in \mathcal{H}, x \in X$. It is also useful to recall that $k$ is a reproducing kernel if and only if there exists a Hilbert (feature) space $\mathcal{F}$ and a (feature) map $\phi : X \to \mathcal{F}$ such that

$$k(x, x') = \langle \phi(x), \phi(x') \rangle, \qquad \forall x, x' \in X, \qquad (7)$$

where $\mathcal{F}$ can be infinite dimensional.

The connection to RKHS is interesting in at least two ways. First, it allows to use results and techniques from the theory of RKHS to analyze random features. Second, it shows that random features can be seen as an approach to derive scalable kernel methods [10]. Indeed, kernel methods have complexity at least quadratic in the number of points, while random features have complexity which is typically linear in the number of points. From this point of view, the intuition behind random features is to relax (7) considering

$$k(x, x') \approx \langle \phi_M(x), \phi_M(x') \rangle, \qquad \forall x, x' \in X. \qquad (8)$$

where $\phi_M$ is finite dimensional.

## 2.2 Computational complexity

If we assume the computation of the feature map $\phi_M(x)$ to have a constant cost, the iteration (3) requires $O(M)$ operations per iteration for $b = 1$, that is $O(Mn)$ for one pass $T = n$. Note that for $b > 1$ each iteration cost $O(Mb)$ but one pass corresponds to $\lceil \frac{n}{b} \rceil$ iterations so that the cost for one pass is again $O(Mn)$. A main advantage of mini-batching is that gradient computations can be easily parallelized. In the multiple pass case, the time complexity after $T$ iterations is $O(MbT)$.
Computing the feature map $\phi_M(x)$ requires to compute $M$ random features. The computation of one random feature does not depend on $n$, but only on the input space $X$. If for example we assume $X \subseteq \mathbb{R}^D$ and consider random features defined as in the previous section, computing $\phi_M(x)$ requires $M$ random projections of $D$ dimensional vectors [6], for a total time complexity of $O(MD)$ for evaluating the feature map at one point. For different input spaces and different types of random features computational cost may differ, see for example Orthogonal Random Features [16] or Fastfood [17] where the cost is reduced from $O(MD)$ to $O(M \log D)$. Note that the analysis presented in his paper holds for random features which are independent, while Orthogonal and Fastfood random features are dependent. Although it should be possible to extend our analysis for Orthogonal and Fastfood random features, further work is needed. To simplify the discussion, in the following we treat the complexity of $\phi_M(x)$ to be $O(M)$.
One of the advantages of random features is that each $\phi_M(x)$ can be computed online at each iteration, preserving $O(MbT)$ as the time complexity of the algorithm (3). Computing $\phi_M(x)$ online also reduces memory requirements. Indeed the space complexity of the algorithm (3) is $O(Mb)$ if the mini-batches are computed in parallel, or $O(M)$ if computed sequentially.

## 2.3 Related approaches

We comment on the connection to related algorithms. Random features are typically used within an empirical risk minimization framework [18]. Results considering convex Lipschitz loss functions and $\ell_\infty$ constraints are given in [19], while [20] considers $\ell_2$ constraints. A ridge regression framework is considered in [8], where it is shown that it is possible to achieve optimal statistical guarantees with a number of random features in the order of $\sqrt{n}$. The combination of random features and gradient methods is less explored. A stochastic coordinate descent approach is considered in [21], see also [22, 23]. A related approach is based on subsampling and is often called Nyström method [24, 25]. Here a shallow network is defined considering a nonlinearity which is a positive definite kernel, and weights chosen as a subset of training set points. This idea can be used within a penalized empirical risk minimization framework [26, 27, 28] but also considering gradient [29, 30] and stochastic gradient [31] techniques. An empirical comparison between Nyström method, random features and full kernel method is given in [23], where the empirical risk minimization problem is solved by block coordinate descent. Note that numerous works have combined stochastic gradient and kernel methods with no random projections approximation [32, 33, 34, 35, 36, 5]. The above list of references is only partial and focusing on papers providing theoretical analysis. In the following, after stating our main results we provide a further quantitative comparison with related results.

# 3 Main Results

In this section, we first discuss our main results under basic assumptions and then more refined results under further conditions.

## 3.1 Worst case results

Our results apply to a general class of random features described by the following assumption.

**Assumption 1.** *Let $(\Omega, \pi)$ be a probability space, $\psi : X \times \Omega \to \mathbb{R}$ and for all $x \in X$,*

$$\phi_M(x) = \frac{1}{\sqrt{M}} \left( \psi(x, \omega_1), \ldots, \psi(x, \omega_M) \right), \qquad (9)$$

*where $\omega_1, \ldots, \omega_M \in \Omega$ are sampled independently according to $\pi$.*

The above class of random features cover all the examples described in section 2.1, as well as many others, see [8, 20] and references therein. Next we introduce the positive definite kernel defined by the above random features. Let $k : X \times X \to \mathbb{R}$ be defined by

$$k(x, x') = \int \psi(x, \omega) \psi(x', \omega) d\pi(\omega), \quad \forall, x, x' \in X.$$

It is easy to check that $k$ is a symmetric and positive definite kernel. To control basic properties of the induced kernel (continuity, boundedness), we require the following assumption, which is again satisfied by the examples described in section 2.1 (see also [8, 20] and references therein).

**Assumption 2.** *The function $\psi$ is continuous and there exists $\kappa \geq 1$ such that $|\psi(x, \omega)| \leq \kappa$ for any $x \in X, \omega \in \Omega$.*

The kernel introduced above allows to compare random feature maps of different size and to express the regularity of the largest function class they induce. In particular, we require a standard assumption in the context of non-parametric regression (see [11]), which consists in assuming a minimum for the expected risk, over the space of functions induced by the kernel.

**Assumption 3.** *If $\mathcal{H}$ is the RKHS with kernel $k$, there exists $f_{\mathcal{H}} \in \mathcal{H}$ such that*

$$\mathcal{E}(f_{\mathcal{H}}) = \inf_{f \in \mathcal{H}} \mathcal{E}(f).$$

To conclude, we need some basic assumption on the data distribution. For all $x \in X$, we denote by $\rho(y|x)$ the conditional probability of $\rho$ and by $\rho_X$ the corresponding marginal probability on $X$. We need a standard moment assumption to derive probabilistic results.

**Assumption 4.** *For any $x \in X$*

$$\int_{\mathcal{Y}} y^{2l} d\rho(y|x) \leq l! B^l p, \qquad \forall l \in \mathbb{N} \tag{10}$$

*for costants $B \in (0, \infty)$ and $p \in (1, \infty)$, $\rho_X$-almost surely.*

The above assumption holds when $y$ is bounded, sub-gaussian or sub-exponential.
The next theorem corresponds to our first main result. Recall that, the *excess risk* for a given estimator $\widehat{f}$ is defined as

$$\mathcal{E}(\widehat{f}) - \mathcal{E}(f_{\mathcal{H}}),$$

and is a standard error measure in statistical machine learning [11, 18]. In the following theorem, we control the *excess risk* of the estimator with respect to the number of points, the number of RF, the step size, the mini-batch size and the number of iterations. We let $\widehat{f}_{t+1} = \langle \widehat{w}_{t+1}, \phi_M(\cdot) \rangle$, with $\widehat{w}_{t+1}$ as in (3).

**Theorem 1.** *Let $n, M \in \mathbb{N}_+$, $\delta \in (0, 1)$ and $t \in [T]$. Under Assumptions 1 to 4, for $b \in [n]$, $\gamma_t = \gamma$ s.t. $\gamma \leq \frac{n}{9T \log \frac{n}{\delta}} \wedge \frac{1}{8(1+\log T)}$, $n \geq 32 \log^2 \frac{2}{\delta}$ and $M \gtrsim \gamma T$ the following holds with probability at least $1 - \delta$:*

$$\mathbb{E}_J \big[ \mathcal{E}(\widehat{f}_{t+1}) \big] - \mathcal{E}(f_{\mathcal{H}}) \lesssim \frac{\gamma}{b} + \left( \frac{\gamma t}{M} + 1 \right) \frac{\gamma t \log \frac{1}{\delta}}{n} + \frac{\log \frac{1}{\delta}}{M} + \frac{1}{\gamma t}. \tag{11}$$

The above theorem bounds the excess risk with a sum of terms controlled by the different parameters. The following corollary shows how these parameters can be chosen to derive finite sample bounds.

**Corollary 1.** *Under the same assumptions of Theorem 1, for one of the following conditions*

*($c_{1.1}$). $b = 1$, $\gamma \simeq \frac{1}{n}$, and $T = n\sqrt{n}$ iterations ($\sqrt{n}$ passes over the data);*

*($c_{1.2}$). $b = 1$, $\gamma \simeq \frac{1}{\sqrt{n}}$, and $T = n$ iterations (1 pass over the data);*

*($c_{1.3}$). $b = \sqrt{n}$, $\gamma \simeq 1$, and $T = \sqrt{n}$ iterations (1 pass over the data);*

*($c_{1.4}$). $b = n$, $\gamma \simeq 1$, and $T = \sqrt{n}$ iterations ($\sqrt{n}$ passes over the data);*

*a number*

$$M = \widetilde{O}(\sqrt{n}) \tag{12}$$

*of random features is sufficient to guarantee with high probability that*

$$\mathbb{E}_J \big[ \mathcal{E}(\widehat{f}_T) \big] - \mathcal{E}(f_{\mathcal{H}}) \lesssim \frac{1}{\sqrt{n}}. \tag{13}$$

The above learning rate is the same achieved by an exact kernel ridge regression (KRR) estimator [11, 37, 38], which has been proved to be optimal in a minimax sense [11] under the same assumptions. Further, the number of random features required to achieve this bound is the same as the kernel ridge regression estimator with random features [8]. Notice that, for the limit case where the number of random features grows to infinity for Corollary 1 under conditions ($c_{1.2}$) and ($c_{1.3}$) we recover the same results for one pass SGD of [39], [40]. In this limit, our results are also related to those in [41]. Here, however, averaging of the iterates is used to achieve larger step-sizes.
Note that conditions ($c_{1.1}$) and ($c_{1.2}$) in the corollary above show that, when no mini-batches are used ($b = 1$) and $\frac{1}{n} \leq \gamma \leq \frac{1}{\sqrt{n}}$, then the step-size $\gamma$ determines the number of passes over the data required for optimal generalization. In particular the number of passes varies from constant, when $\gamma = \frac{1}{\sqrt{n}}$, to $\sqrt{n}$, when $\gamma = \frac{1}{n}$. In order to increase the step-size over $\frac{1}{\sqrt{n}}$ the algorithm needs to be run with mini-batches. The step-size can then be increased up to a constant if $b$ is chosen equal to $\sqrt{n}$ (condition ($c_{1.3}$)), requiring the same number of passes over the data of the setting ($c_{1.2}$). Interestingly condition ($c_{1.4}$) shows that increasing the mini-batch size over $\sqrt{n}$ does not allow to take larger step-sizes, while it seems to increase the number of passes over the data required to reach optimality.
We now compare the time complexity of algorithm (3) with some closely related methods which

achieve the same optimal rate of $\frac{1}{\sqrt{n}}$. Computing the classical KRR estimator [11] has a complexity of roughly $O(n^3)$ in time and $O(n^2)$ in memory. Lowering this computational cost is possible with random projection techniques. Both random features and Nyström method on KRR [8, 26] lower the time complexity to $O(n^2)$ and the memory complexity to $O(n\sqrt{n})$ preserving the statistical accuracy. The same time complexity is achieved by stochastic gradient method solving the full kernel method [33, 36], but with the higher space complexity of $O(n^2)$. The combination of the stochastic gradient iteration, random features and mini-batches allows our algorithm to achieve a complexity of $O(n\sqrt{n})$ in time and $O(n)$ in space for certain choices of the free parameters (like $(c_{1.2})$ and $(c_{1.3})$). Note that these time and memory complexity are lower with respect to those of stochastic gradient with mini-batches and Nyström approximation which are $O(n^2)$ and $O(n)$ respectively [31]. A method with similar complexity to SGD with RF is FALKON [30]. This method has indeed a time complexity of $O(n\sqrt{n}\log(n))$ and $O(n)$ space complexity. This method blends together Nyström approximation, a sketched preconditioner and conjugate gradient.

## 3.2 Refined analysis and fast rates

We next discuss how the above results can be refined under an additional regularity assumption. We need some preliminary definitions. Let $\mathcal{H}$ be the RKHS defined by $k$, and $L : L^2(X, \rho_X) \to L^2(X, \rho_X)$ the integral operator

$$Lf(x) = \int k(x, x')f(x')d\rho_X(x'), \qquad \forall f \in L^2(X, \rho_X), x \in X,$$

where $L^2(X, \rho_X) = \{f : X \to \mathbb{R} : \|f\|_\rho^2 = \int |f|^2 d\rho_X < \infty\}$. The above operator is symmetric and positive definite. Moreover, Assumption 1 ensures that the kernel is bounded, which in turn ensures $L$ is trace class, hence compact [18].

**Assumption 5.** *For any $\lambda > 0$, define the effective dimension as $\mathcal{N}(\lambda) = \mathrm{Tr}((L + \lambda I)^{-1}L)$, and assume there exist $Q > 0$ and $\alpha \in [0, 1]$ such that*

$$\mathcal{N}(\lambda) \leq Q^2\lambda^{-\alpha}. \tag{14}$$

*Moreover , assume there exists $r \geq 1/2$ and $g \in L^2(X, \rho_X)$ such that*

$$f_{\mathcal{H}}(x) = (L^r g)(x). \tag{15}$$

Condition (14) describes the *capacity/complexity* of the RKHS $\mathcal{H}$ and the measure $\rho$. It is equivalent to classic entropy/covering number conditions, see e.g. [18]. The case $\alpha = 1$ corresponds to making no assumptions on the kernel, and reduces to the worst case analysis in the previous section. The smaller is $\alpha$ the more stringent is the capacity condition. A classic example is considering $X = \mathbb{R}^D$ with $d\rho_X(x) = p(x)dx$, where $p$ is a probability density, strictly positive and bounded away from zero, and $\mathcal{H}$ to be a Sobolev space with smoothness $s > D/2$. Indeed, in this case $\alpha = D/2s$ and classical nonparametric statistics assumptions are recovered as a special case. Note that in particular the worst case is $s = D/2$. Condition (15) is a regularity condition commonly used in approximation theory to control the bias of the estimator [42].

The following theorem is a refined version of Theorem 1 where we also consider the above *capacity* condition (Assumption 5).

**Theorem 2.** *Let $n, M \in \mathbb{N}_+$, $\delta \in (0, 1)$ and $t \in [T]$, under Assumptions 1 to 4, for $b \in [n]$, $\gamma_t = \gamma$ s.t. $\gamma \leq \frac{n}{9T\log\frac{n}{\delta}} \wedge \frac{1}{8(1+\log T)}$, $n \geq 32\log^2\frac{2}{\delta}$ and $M \gtrsim \gamma T$ the following holds with high probability:*

$$\mathbb{E}_J\left[\mathcal{E}(\widehat{f}_{t+1})\right] - \mathcal{E}(f_{\mathcal{H}}) \lesssim \frac{\gamma}{b} + \left(\frac{\gamma t}{M} + 1\right)\frac{\mathcal{N}\left(\frac{1}{\gamma t}\right)\log\frac{1}{\delta}}{n} + \frac{\mathcal{N}\left(\frac{1}{\gamma t}\right)^{2r-1}\log\frac{1}{\delta}}{M(\gamma t)^{2r-1}} + \left(\frac{1}{\gamma t}\right)^{2r}. \tag{16}$$

The main difference is the presence of the effective dimension providing a sharper control of the stability of the considered estimator. As before, explicit learning bounds can be derived considering different parameter settings.

**Corollary 2.** *Under the same assumptions of Theorem 2, for one of the following conditions*

$(c_{2.1})$. *$b = 1$, $\gamma \simeq n^{-1}$, and $T = n^{\frac{2r+\alpha+1}{2r+\alpha}}$ iterations ($n^{\frac{1}{2r+\alpha}}$ passes over the data);*

($c_{2.2}$). $b = 1$, $\gamma \simeq n^{-\frac{2r}{2r+\alpha}}$, and $T = n^{\frac{2r+1}{2r+\alpha}}$ iterations ($n^{\frac{1-\alpha}{2r+\alpha}}$ passes over the data);

($c_{2.3}$). $b = n^{\frac{2r}{2r+\alpha}}$, $\gamma \simeq 1$, and $T = n^{\frac{1}{2r+\alpha}}$ iterations ($n^{\frac{1-\alpha}{2r+\alpha}}$ passes over the data);

($c_{2.4}$). $b = n$, $\gamma \simeq 1$, and $T = n^{\frac{1}{2r+\alpha}}$ iterations ($n^{\frac{1}{2r+\alpha}}$ passes over the data);

a number

$$M = \widetilde{O}(n^{\frac{1+\alpha(2r-1)}{2r+\alpha}}) \tag{17}$$

of random features suffies to guarantee with high probability that

$$\mathbb{E}_J\big[\mathcal{E}(\widehat{w}_T)\big] - \mathcal{E}(f_\mathcal{H}) \lesssim n^{-\frac{2r}{2r+\alpha}}. \tag{18}$$

The corollary above shows that multi-pass SGD achieves a learning rate that is the same as kernel ridge regression under the regularity assumption 5 and is again minimax optimal (see [11]). Moreover, we obtain the minimax optimal rate with the same number of random features required for ridge regression with random features [8] under the same assumptions. Finally, when the number of random features goes to infinity we also recover the results for the infinite dimensional case of the single-pass and multiple pass stochastic gradient method [33].

It is worth noting that, under the additional regularity assumption 5, the number of both random features and passes over the data sufficient for optimal learning rates increase with respect to the one required in the worst case (see Corollary 1). The same effect occurs in the context of ridge regression with random features as noted in [8]. In this latter paper, it is observed that this issue tackled can be using more refined, possibly more costly, sampling schemes [20].

Finally, we present a general result from which all our previous results follow as special cases. We consider a more general setting where we allow decreasing step-sizes.

**Theorem 3.** *Let $n, M, T \in \mathbb{N}$, $b \in [n]$ and $\gamma > 0$. Let $\delta \in (0,1)$ and $\widehat{w}_{t+1}$ be the estimator in Eq. (3) with $\gamma_t = \gamma \kappa^{-2} t^{-\theta}$ and $\theta \in [0,1[$. Under Assumptions 1 to 4, when $n \geq 32 \log^2 \frac{2}{\delta}$ and*

$$\gamma \leq \frac{n}{9T^{1-\theta} \log \frac{n}{\delta}} \wedge \begin{cases} \frac{\theta \wedge (1-\theta)}{7} & \theta \in ]0,1[ \\ \frac{1}{8(1+\log T)} & otherwise, \end{cases} \tag{19}$$

*moreover*

$$M \geq \big(4 + 18\gamma T^{1-\theta}\big) \log \frac{12\gamma T^{1-\theta}}{\delta}, \tag{20}$$

*then, for any $t \in [T]$ the following holds with probability at least $1 - 9\delta$*

$$\mathbb{E}_J\big[\mathcal{E}(\widehat{w}_{t+1})\big] - \inf_{w \in \mathcal{F}} \mathcal{E}(w) \leq c_1 \frac{\gamma}{b t^{\min(\theta, 1-\theta)}} (\log t \vee 1) \tag{21}$$

$$+ \left(c_2 + c_3 \frac{1}{M} \log \frac{M}{\delta} \left(\gamma t^{1-\theta} \vee 1\right)\right) \frac{\mathcal{N}\left(\frac{\kappa^2}{\gamma t^{1-\theta}}\right)}{n} \left(\log^2(t) \vee 1\right) \log^2 \frac{4}{\delta} \tag{22}$$

$$+ c_4 \left(\frac{\mathcal{N}(\frac{\kappa^2}{\gamma t^{1-\theta}})^{2r-1} \log \frac{2}{\delta}}{M(\gamma t^{1-\theta} \kappa^{-2})^{2r-1}} \log^{2-2r}\left(11\gamma t^{1-\theta}\right) + \left(\frac{1}{\gamma t^{1-\theta}}\right)^{2r}\right), \tag{23}$$

*with $c_1, c_2, c_3, c_4$ constants which do not depend on $b, \gamma, n, t, M, \delta$.*

We note that as the number of random features $M$ goes to infinity, we recover the same bound of [33] for decreasing step-sizes. Moreover, the above theorem shows that there is no apparent gain in using a decreasing stepsize (i.e. $\theta > 0$) with respect to the regimes identified in Corollaries 1 and 2.

### 3.3 Sketch of the Proof

In this section, we sketch the main ideas in the proof. We relate $\widehat{f}_t$ and $f_\mathcal{H}$ introducing several intermediate functions. In particular, the following iterations are useful,

$$\widehat{v}_1 = 0; \qquad \widehat{v}_{t+1} = \widehat{v}_t - \gamma_t \frac{1}{n} \sum_{i=1}^n \big(\langle \widehat{v}_t, \phi_M(x_i)\rangle - y_i\big) \phi_M(x_i), \qquad \forall t \in [T]. \tag{24}$$

$$\widetilde{v}_1 = 0; \qquad \widetilde{v}_{t+1} = \widetilde{v}_t - \gamma_t \int_X \big(\langle \widetilde{v}_t, \phi_M(x)\rangle - y\big) \phi_M(x) d\rho(x,y), \qquad \forall t \in [T]. \tag{25}$$

$$v_1 = 0; \qquad v_{t+1} = v_t - \gamma_t \int_X \big(\langle v_t, \phi_M(x)\rangle - f_\mathcal{H}(x)\big) \phi_M(x) d\rho_X(x), \qquad \forall t \in [T]. \tag{26}$$

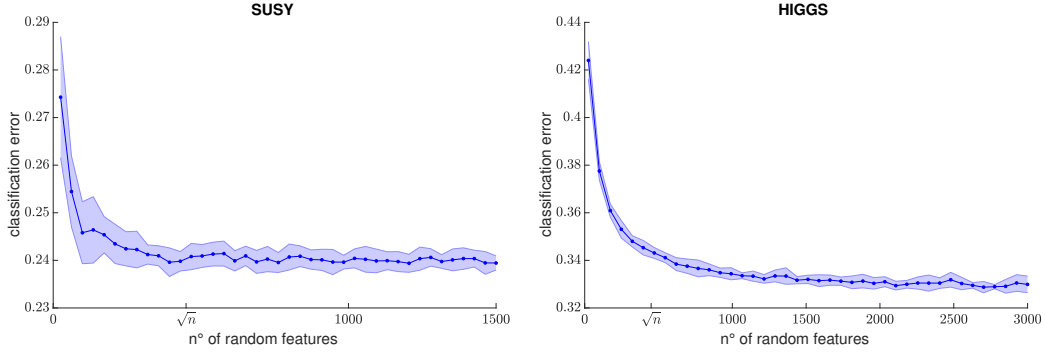

Figure 1: Classification error of SUSY (left) and HIGGS (right) datasets as the $n^o$ of random features varies

Further, we let

$$\widetilde{u}_\lambda = \operatorname*{argmin}_{u \in \mathbb{R}^M} \int_X \left( \langle u, \phi_M(x) \rangle - f_{\mathcal{H}}(x) \right)^2 d\rho_X(x) + \lambda \|u\|^2, \quad \lambda > 0, \tag{27}$$

$$u_\lambda = \operatorname*{argmin}_{u \in \mathcal{F}} \int_X \left( \langle u, \phi(x) \rangle - y \right)^2 d\rho(x,y) + \lambda \|u\|^2, \quad \lambda > 0, \tag{28}$$

where $(\mathcal{F}, \phi)$ are feature space and feature map associated to the kernel $k$. The first three vectors are defined by the random features and can be seen as an empirical and population batch gradient descent iterations. The last two vectors can be seen as a population version of ridge regression defined by the random features and the feature map $\phi$, respectively.

Since the above objects (24), (25), (26), (27), (28) belong to different spaces, instead of comparing them directly we compare the functions in $L^2(X, \rho_X)$ associated to them, letting

$$\widehat{g}_t = \langle \widehat{v}_t, \phi_M(\cdot) \rangle , \quad \widetilde{g}_t = \langle \widetilde{v}_t, \phi_M(\cdot) \rangle , \quad g_t = \langle v_t, \phi_M(\cdot) \rangle , \quad \widetilde{g}_\lambda = \langle \widetilde{u}_\lambda, \phi_M(\cdot) \rangle , \quad g_\lambda = \langle u_\lambda, \phi(\cdot) \rangle .$$

Since it is well known [11] that

$$\mathcal{E}(f) - \mathcal{E}(f_{\mathcal{H}}) = \|f - f_{\mathcal{H}}\|_\rho^2,$$

we than can consider the following decomposition

$$\widehat{f}_t - f_{\mathcal{H}} = \widehat{f}_t - \widehat{g}_t \tag{29}$$
$$+ \widehat{g}_t - \widetilde{g}_t \tag{30}$$
$$+ \widetilde{g}_t - g_t \tag{31}$$
$$+ g_t - \widetilde{g}_\lambda \tag{32}$$
$$+ \widetilde{g}_\lambda - g_\lambda \tag{33}$$
$$+ g_\lambda - f_{\mathcal{H}}. \tag{34}$$

The first two terms control how SGD deviates from the batch gradient descent and the effect of noise and sampling. They are studied in Lemma 1, 2, 3, 4, 5, 6 in the Appendix, borrowing and adapting ideas from [33, 36, 8]. The following terms account for the approximation properties of random features and the bias of the algorithm. Here the basic idea and novel result is the study of how the population gradient decent and ridge regression are related (32) (Lemma 9 in the Appendix). Then, results from the the analysis of ridge regression with random features are used [8].

## 4 Experiments

We study the behavior of the SGD with RF algorithm on subsets of $n = 2 \times 10^5$ points of the SUSY [3] and HIGGS [4] datasets [43]. The measures we show in the following experiments are an average over 10 repetitions of the algorithm. Further, we consider random Fourier features that

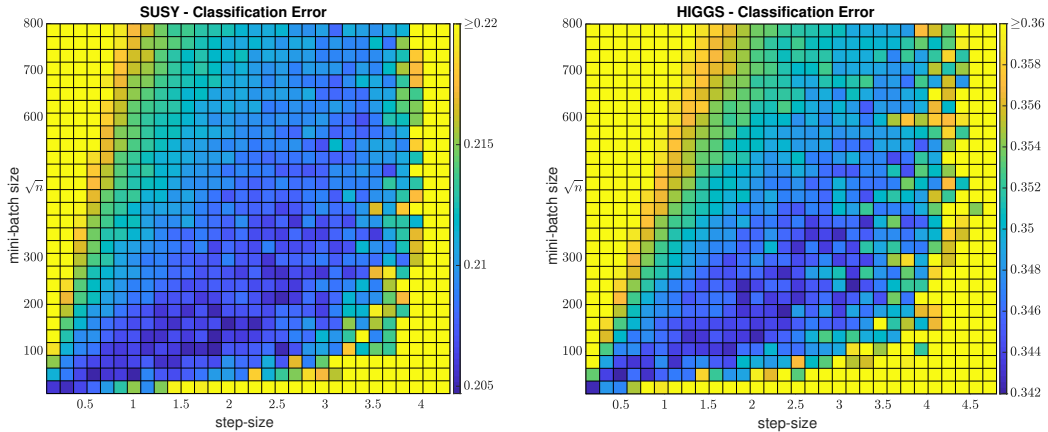

Figure 2: Classification error of SUSY (left) and HIGGS (right) datasets as step-seize and mini-batch size vary

are known to approximate translation invariant kernels [6]. We use random features of the form $\psi(x,\omega) = \cos(w^T x + q)$, with $\omega := (w, q)$, $w$ sampled according to the normal distribution and $q$ sampled uniformly at random between 0 and $2\pi$. Note that the random features defined this way satisfy Assumption 2.

Our theoretical analysis suggests that only a number of RF of the order of $\sqrt{n}$ suffices to gain optimal learning properties. Hence we study how the number of RF affect the accuracy of the algorithm on test sets of $10^5$ points. In Figure 3.3 we show the classification error after 5 passes over the data of SGD with RF as the number of RF increases, with a fixed batch size of $\sqrt{n}$ and a step-size of 1. We can observe that over a certain threshold of the order of $\sqrt{n}$, increasing the number of RF does not improve the accuracy, confirming what our theoretical results suggest.

Further, theory suggests that the step-size can be increased as the mini-batch size increases to reach an optimal accuracy, and that after a mini-batch size of the order of $\sqrt{n}$ more than 1 pass over the data is required to reach the same accuracy. We show in Figure 2 the classification error of SGD with RF after 1 pass over the data, with a fixed number of random features $\sqrt{n}$, as mini-batch size and step-size vary, on test sets of $10^5$ points. As suggested by theory, to reach the lowest error as the mini-batch size grows the step-size needs to grow as well. Further for mini-batch sizes bigger that $\sqrt{n}$ the lowest error can not be reached in only 1 pass even if increasing the step-size.

## 5    Conclusions

In this paper we investigate the combination of sketching and stochastic techniques in the context of non-parametric regression. In particular we studied the statistical and computational properties of the estimator defined by stochastic gradient descent with multiple passes, mini-batches and random features. We proved that the estimator achieves optimal statistical properties with a number of random features in the order of $\sqrt{n}$ (with $n$ the number of examples). Moreover we analyzed possible trade-offs between the number of passes, the step and the dimension of the mini-batches showing that there exist different configurations which achieve the same optimal statistical guarantees, with different computational impacts.

Our work can be extended in several ways: First, (a) we can study the effect of combining random features with accelerated/averaged stochastic techniques as [32]. Second, (b) we can extend our analysis to consider more refined assumptions, generalizing [35] to SGD with random features. Additionally, (c) we can study the statistical properties of the considered estimator in the context of classification with the goal of showing fast decay of the classification error, as in [34]. Moreover, (d) we can apply the proposed method in the more general context of least squares frameworks for multitask learning [44, 45] or structured prediction [46, 47, 48], with the goal of obtaining faster algorithms, while retaining strong statistical guarantees. Finally, (e) to integrate our analysis with more refined methods to select the random features analogously to [49, 50] in the context of column sampling.

**Acknowledgments.**
This material is based upon work supported by the Center for Brains, Minds and Machines (CBMM), funded by NSF STC award CCF-1231216, and the Italian Institute of Technology. We gratefully acknowledge the support of NVIDIA Corporation for the donation of the Titan Xp GPUs and the Tesla k40 GPU used for this research. L. R. acknowledges the support of the AFOSR projects FA9550-17-1-0390 and BAA-AFRL-AFOSR-2016-0007 (European Office of Aerospace Research and Development), and the EU H2020-MSCA-RISE project NoMADS - DLV-777826. A. R. acknowledges the support of the European Research Council (grant SEQUOIA 724063).

## Footnotes

[2]For all $x_1, \ldots, x_n$ the matrix with entries $k(x_i, x_j), i, j = 1, \ldots, n$ is positive semi-definite.

[3] https://archive.ics.uci.edu/ml/datasets/SUSY

[4] https://archive.ics.uci.edu/ml/datasets/HIGGS

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
