[Supplementary Material · SGD-RF_supplementary.pdf]

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

# A Appendix

We start recalling some definitions and define some new operators.

## A.1 Preliminary definitions

Let $\mathcal{F}$ be the feature space corresponding to the kernel $k$ given in Assumption 2.

Given $\phi\colon X \to \mathcal{F}$ (feature map), we define the operator $S\colon \mathcal{F} \to L^2(X, \rho_X)$ as

$$(Sw)(\cdot) = \langle w, \phi(\cdot)\rangle_{\mathcal{F}}, \qquad \forall w \in \mathcal{F}. \tag{35}$$

If $S^*$ is the adjoint operator of $S$, we let $C\colon \mathcal{F} \to \mathcal{F}$ be the linear operator $C = S^*S$, which can be written as

$$C = \int_X \phi(x) \otimes \phi(x) d\rho_X(x). \tag{36}$$

We also define the linear operator $L\colon L^2(X, \rho_X) \to L^2(X, \rho_X)$ such that $L = SS^*$, that can be represented as

$$(Lf)(\cdot) = \int_X \langle \phi(x), \phi(\cdot)\rangle_{\mathcal{F}}\, f(x) d\rho_X(x), \qquad \forall f \in L^2(X, \rho_X). \tag{37}$$

We now define the analog of the previous operators where we use the feature map $\phi_M$ instead of $\phi$. We have $S_M\colon \mathbb{R}^M \to L^2(X, \rho_X)$ defined as

$$(S_M v)(\cdot) = \langle v, \phi_M(\cdot)\rangle_{\mathbb{R}^M}, \qquad \forall v \in \mathbb{R}^M, \tag{38}$$

together with $C_M\colon \mathbb{R}^M \to \mathbb{R}^M$ and $L_M\colon L^2(X, \rho_X) \to L^2(X, \rho_X)$ defined as $C_M = S_M^*S_M$ and $L_M = S_M S_M^*$ respectively.

We also define the empirical counterpart of the previous operators. The operator $\widehat{S}_M\colon \mathbb{R}^M \to \mathbb{R}^n$ is defined as,

$$\widehat{S}_M^\top = \frac{1}{\sqrt{n}} \left(\phi_M(x_1), \ldots, \phi_M(x_n)\right), \tag{39}$$

and with $\widehat{C}_M\colon \mathbb{R}^M \to \mathbb{R}^M$ and $\widehat{L}_M\colon \mathbb{R}^n \to \mathbb{R}^n$ are defined as $\widehat{C}_M = \widehat{S}_M^\top \widehat{S}_M$ and $\widehat{L}_M = \widehat{S}_M \widehat{S}_M^\top$, respectively.

**Remark 1** (from [51, 52]). *Let $P : L^2(X, \rho_X) \to L^2(X, \rho_X)$ be the projection operator whose range is the closure of the range of $L$. Let $f_\rho : X \to \mathbb{R}$ be defined as*

$$f_\rho = \int y d\rho(y|x).$$

*If there exists $f_{\mathcal{H}} \in \mathcal{H}$ such that*

$$\inf_{f \in \mathcal{H}} \mathcal{E}(f) = \mathcal{E}(f_{\mathcal{H}}),$$

*then*

$$Pf_\rho = Sf_{\mathcal{H}},$$

*or equivalently, there exists $g \in L^2(X, \rho_X)$ such that*

$$Pf_\rho = L^r g,$$

*with $1/2 \leq r \leq 1$. In particular, we have $R := \|f_{\mathcal{H}}\|_{\mathcal{H}} = \|g\|_{L^2(X, \rho_X)}$.*

With the operators introduced above and Remark 1, we can rewrite the auxiliary objects (24), (25), (26), (27), (28) respectively as

$$\widehat{v}_1 = 0; \qquad \widehat{v}_{t+1} = (I - \gamma_t \widehat{C}_M)\widehat{v}_t + \gamma_t \widehat{S}_M^* \widehat{y}, \qquad \forall t \in [T], \tag{40}$$

$$\widetilde{v}_1 = 0; \qquad \widetilde{v}_{t+1} = (I - \gamma_t C_M)\widetilde{v}_t + \gamma_t S_M^* f_\rho, \qquad \forall t \in [T], \tag{41}$$

$$v_1 = 0; \qquad v_{t+1} = (I - \gamma_t C_M)v_t + \gamma_t S_M^* P f_\rho, \qquad \forall t \in [T]. \tag{42}$$

where $\widehat{y} = n^{-1/2}(y_1, \ldots, y_n)$, and

$$\widetilde{u}_\lambda = S_M^* L_{M,\lambda}^{-1} P f_\rho \tag{43}$$

$$u_\lambda = S^* L_\lambda^{-1} P f_\rho \tag{44}$$

By a simple induction argument the three sequences can be written as

$$\widehat{v}_{t+1} = \sum_{i=1}^t \gamma_i \prod_{k=i+1}^t (I - \gamma_k \widehat{C}_M) \widehat{S}_M^* \widehat{y} \tag{45}$$

$$\widetilde{v}_{t+1} = \sum_{i=1}^t \gamma_i \prod_{k=i+1}^t (I - \gamma_k \widehat{C}_M) S_M^* f_\rho \tag{46}$$

$$v_{t+1} = \sum_{i=1}^t \gamma_i \prod_{k=i+1}^t (I - \gamma_k C_M) S_M^* P f_\rho \tag{47}$$

## A.2 Error decomposition

We can now rewrite the error decomposition of $\widehat{f}_t - f_{\mathcal{H}}$ using the operators introduced above as

$$S_M \widehat{w}_t - P f_\rho = S_M \widehat{w}_t - S_M \widehat{v}_t \tag{48}$$

$$+ S_M \widehat{v}_t - S_M \widetilde{v}_t \tag{49}$$

$$+ S_M \widetilde{v}_t - S_M v_t \tag{50}$$

$$+ S_M v_t - L_M L_{M,\lambda}^{-1} P f_\rho \tag{51}$$

$$+ L_M L_{M,\lambda}^{-1} P f_\rho - L L_\lambda^{-1} P f_\rho \tag{52}$$

$$+ L L_\lambda^{-1} P f_\rho - P f_\rho. \tag{53}$$

## A.3 Lemmas

The first three lemmas we present are some technical lemmas used when bounding the first three terms (48), (49), (50) of the error decomposition.

**Lemma 1.** *Under Assumption 2 the following holds for any $t, M, n \in \mathbb{N}$*

$$\|\widetilde{v}_t - v_t\| = 0 \quad a.s. \tag{54}$$

*Proof.* Given (46), (47) and defining $A_{Mt} = \sum_{i=1}^t \gamma_i \prod_{k=i+1}^t (I - \gamma_k C_M)$, we can write

$$\|\widetilde{v}_t - v_t\| = \|A_{Mt} S_M^* (I - P) f_\rho\| \leq \|A_{Mt}\| \|S_M^* (I - P)\| \|f_\rho\|. \tag{55}$$

Under Assumption 2, by Lemma 2 of [8], we have $\|S_M^* (I - P)\| = 0$, which completes the proof. □

**Lemma 2.** *Let $M \in \mathbb{N}$. Under Assumption 2 and 3, let $\gamma_t \kappa^2 \leq 1$, $\delta \in ]0, 1]$, the following holds with probability $1 - \delta$ for all $t \in [T]$*

$$\|\widetilde{v}_{t+1}\| \leq 2 R \kappa^{2r-1} \left(1 + \sqrt{\frac{9\kappa^2}{M} \log \frac{M}{\delta}} \max\left(\left(\sum_{i=1}^t \gamma_t\right)^{\frac{1}{2}}, \kappa^{-1}\right)\right). \tag{56}$$

*Proof.* Considering (41) (42), we can write

$$\|\widetilde{v}_{t+1}\| \leq \|\widetilde{v}_{t+1} - v_{t+1}\| + \|v_{t+1}\| = \|v_{t+1}\|, \tag{57}$$

where in the last equality we used the result from Lemma 1. Using Assumption 3 (see also Remark 1), we derive

$$\|v_{t+1}\| = \left\|\sum_{i=1}^t \gamma_i S_M^* \prod_{k=i+1}^t (I - \gamma_k L_M) P f_\rho\right\| \leq R \left\|\sum_{i=1}^t \gamma_i S_M^* \prod_{k=i+1}^t (I - \gamma_k L_M) L^r\right\| \tag{58}$$

Define $Q_{Mt} = \sum_{i=1}^t \gamma_i S_M^* \prod_{k=i+1}^t (I - \gamma_k L_M)$. Note that $\|L^{r-\frac{1}{2}}\| \leq \kappa^{2r-1}$ for $r \geq \frac{1}{2}$ and that $\|L_{M,\eta}^{-1/2} L^{1/2}\| \leq 2$ holds with probability $1 - \delta$ when $\frac{9\kappa^2}{M} \log \frac{M}{\delta} \leq \eta \leq \|L\|$ (see Lemma 5 in [26]).

Moreover, when $\eta \geq \|L\|$, we have that $\|L_{M,\eta}^{-1/2}L^{1/2}\| \leq \eta^{-1/2}\|L^{1/2}\| \leq 1$. So $\|L_{M,\eta}^{-1/2}L^{1/2}\| \leq 2$ with probability $1 - \delta$, when

$$\frac{9\kappa^2}{M}\log\frac{M}{\delta} \leq \eta. \tag{59}$$

So when (59) holds, with probability $1 - \delta$ we can write

$$
\begin{aligned}
R\|Q_{Mt}L^r\| &= R\|Q_{Mt}L_{M,\eta}^{\frac{1}{2}}L_{M,\eta}^{-\frac{1}{2}}L^{\frac{1}{2}}L^{r-\frac{1}{2}}\| \\
&\leq R\|Q_{Mt}L_{M,\eta}^{\frac{1}{2}}\|\|L_{M,\eta}^{-\frac{1}{2}}L^{\frac{1}{2}}\|\|L^{r-\frac{1}{2}}\| \\
&\leq 2R\kappa^{2r-1}\|Q_{Mt}L_{M,\eta}^{\frac{1}{2}}\| \\
&\leq 2R\kappa^{2r-1}\left(\|Q_{Mt}L_M^{\frac{1}{2}}\| + \eta^{\frac{1}{2}}\|Q_{Mt}\|\right).
\end{aligned} \tag{60}
$$

Now note that for any $a \in [0, 1/2]$,

$$\|Q_{Mt}L_M^a\| \leq \max\left(\kappa^{2a-1}, \left(\sum_{i=1}^{t}\gamma_i\right)^{\frac{1}{2}-a}\right) \tag{61}$$

(see Lemma B.10(i) in [36] or Lemma 16 of [33]). We use (61) with $a = \frac{1}{2}$ and $a = 0$ to bound $\|Q_{Mt}L_M^{1/2}\|$ and $\|Q_{Mt}\|$ respectively and plug the results in (60). To complete the proof we take $\eta = \frac{9\kappa^2}{M}\log\frac{M}{\delta}$. $\qquad\square$

**Lemma 3.** *Let* $\lambda > 0$, $R \in \mathbb{N}$ *and* $\delta \in (0,1)$. *Let* $\zeta_1, \ldots, \zeta_R$ *be i.i.d. random vectors bounded by* $\kappa > 0$. *Denote with* $Q_R = \frac{1}{R}\sum_{j=1}^{R}\zeta_j \otimes \zeta_j$ *and by* $Q$ *the expectation of* $Q_R$. *Then, for any* $\lambda \geq \frac{9\kappa^2}{R}\log\frac{R}{\delta}$, *we have*

$$\|(Q_R + \lambda I)^{-1/2}(Q + \lambda I)^{1/2}\|^2 \leq 2.$$

*Proof.* This lemma is a more refined version of a result in [53]. When $\|Q\| \geq \lambda \geq \frac{9\kappa^2}{R}\log\frac{R}{\delta}$, by combining Prop. 8 of [8], with Prop. 6 and in particular Rem. 10 point 2 of the same paper, we have that $\|(Q_R + \lambda I)^{-1/2}(Q + \lambda I)^{1/2}\| \leq 2$, with probability at least $1 - \delta$. To cover the case $\lambda > \|Q\|$, note that

$$\|(Q_R + \lambda I)^{-1/2}(Q + \lambda I)^{1/2}\| \leq (\|Q\|^{1/2} + \lambda^{1/2})/\lambda^{1/2}.$$

When $\lambda > \|Q\|$, we have that

$$\|(Q_R + \lambda I)^{-1/2}(Q + \lambda I)^{1/2}\| \leq \sup_{\lambda > \|Q\|}(\|Q\|^{1/2} + \lambda^{1/2})/\lambda^{1/2} \leq 2.$$

$\qquad\square$

We need the following technical lemma that complements Proposition 10 of [8] when $\lambda \geq \|L\|$, and that we will need for the proof of Lemma 6.

**Lemma 4.** *Let* $M \in \mathbb{N}$ *and* $\delta \in (0,1]$. *For any* $\lambda > 0$ *such that*

$$M \geq \left(4 + \frac{18\kappa^2}{\lambda}\right)\log\frac{12\kappa^2}{\lambda\delta},$$

*the following holds with probability* $1 - \delta$

$$\mathcal{N}_M(\lambda) := \int_X \|(L_M + \lambda I)^{-\frac{1}{2}}\phi_M(x)\|^2 d\rho_X(x) \leq \max\left(2.55, \frac{2\kappa^2}{\|L\|}\right)\mathcal{N}(\lambda).$$

*Proof.* First of all note that

$$\mathcal{N}_M(\lambda) := \int_X \|(L_M + \lambda I)^{-\frac{1}{2}}\phi_M(x)\|^2 d\rho_X(x) = \text{Tr}(L_{M,\lambda}^{-\frac{1}{2}}L_M L_{M,\lambda}^{-\frac{1}{2}}) = \text{Tr}(L_{M,\lambda}^{-1}L_M).$$

Now consider the case when $\lambda \le \|L\|$. By applying Proposition 10 of [8] we have that under the required condition on $M$, the following holds with probability at least $1 - \delta$

$$\mathcal{N}_M(\lambda) \le 2.55 \mathcal{N}(\lambda).$$

For the case $\lambda > \|L\|$, note that $\mathrm{Tr}(AA_\lambda^{-1})$ satisfies the following inequality for any trace class positive linear operator $A$ with trace bounded by $\kappa^2$ and $\lambda > 0$,

$$\frac{\|A\|}{\|A\| + \lambda} \le \mathrm{Tr}(AA_\lambda^{-1}) \le \frac{\mathrm{Tr}(A)}{\lambda}.$$

So, when $\lambda > \|L\|$, since $\mathcal{N}_M(\lambda) = \mathrm{Tr}(C_M C_{M\lambda}^{-1})$ and $\mathcal{N}(\lambda) = \mathrm{Tr}(LL_\lambda^{-1})$, and both $L$ and $\widehat{C}_M$ have trace bounded by $\kappa^2$, we have $\mathcal{N}_M(\lambda) \le \frac{\kappa^2}{\lambda}$ and $\mathcal{N}(\lambda) \ge \frac{\|L\|}{\|L\|+\lambda}$. So by selecting $q = \frac{\kappa^2(\|L\|+\lambda)}{\lambda\|L\|}$, we have

$$\mathcal{N}_M(\lambda) \le \frac{\kappa^2}{\lambda} = q \frac{\|L\|}{\|L\| + \lambda} \le q\mathcal{N}(\lambda).$$

Finally note that

$$q \le \sup_{\lambda > \|L\|} \frac{\kappa^2(\|L\| + \lambda)}{\lambda\|L\|} \le 2\frac{\kappa^2}{\|L\|}.$$

$\square$

We now start bounding the different parts of the error decomposition. The next two lemmas bound the first two terms (48), (49). To bound these we require the above lemmas and adapting ideas from [33, 36, 8].

**Lemma 5.** *Under Assumption 2 and 4, let $\delta \in ]0,1[$, $n \ge 32 \log^2 \frac{2}{\delta}$, and $\gamma_t = \gamma\kappa^{-2}t^{-\theta}$ for all $t \in [T]$, with $\theta \in [0,1[$ and $\gamma$ such that*

$$0 < \gamma \le \frac{t^{\min(\theta, 1-\theta)}}{8(\log t + 1)}, \qquad \forall t \in [T]. \tag{62}$$

*When*

$$\frac{1}{\gamma t^{1-\theta}} \ge \frac{9}{n} \log \frac{n}{\delta} \tag{63}$$

*for all $t \in [T]$, with probability at least $1 - 2\delta$,*

$$\mathbb{E}_{\mathbf{J}}\|S_M(\widehat{w}_{t+1} - \widehat{v}_{t+1})\|^2 \le \frac{208Bp}{(1-\theta)b}\left(\gamma t^{-\min(\theta, 1-\theta)}\right)(\log t \vee 1). \tag{64}$$

*Proof.* The proof is derived by applying Proposition 6 in [33] with $\gamma$ satisfying condition (62), $\lambda = \frac{1}{\gamma_t t}$, $\delta_2 = \delta_3 = \delta$, and some changes that we now describe. Instead of the stochastic iteration $w_t$ and the batch gradient iteration $\nu_t$ as defined in [33] we consider (3) and (40) respectively, as well as the operators $S_M, C_M, L_M, \widehat{S}_M, \widehat{C}_M, \widehat{L}_M$ defined in Section 2 instead of $S_\rho, \mathcal{T}_\rho, \mathcal{L}_\rho, S_\mathbf{x}, \mathcal{T}_\mathbf{x}, \mathcal{L}_\mathbf{x}$ defined in [33]. Instead of assuming that exists a $\kappa \ge 1$ for which $\langle x, x'\rangle \le \kappa^2, \forall x, x' \in X$ we have Assumption 2 which implies the same $\kappa^2$ upper bound of the operators used in the proof. To apply this version of Proposition 6 note that their Equation (63) is satisfied by Lemma 25 of [33], while their Equation (47) is satisfied by our Lemma 3, from which we obtain the condition (63). $\square$

**Lemma 6.** *Under Assumptions 2, 4 and 3, let $\delta \in ]0,1[$ and $\gamma_t = \gamma\kappa^{-2}t^{-\theta}$ for all $t \in [T]$, with $\gamma \in ]0,1]$ and $\theta \in [0,1[$. When*

$$M \ge \left(4 + 18\gamma t^{1-\theta}\right)\log \frac{12\gamma t^{1-\theta}}{\delta}, \tag{65}$$

*for all $t \in [T]$ with probability at least $1 - 3\delta$*

$$\|S_M(\widehat{v}_{t+1} - \widetilde{v}_{t+1})\| \le 4\left(R\kappa^{2r}\left(1 + \sqrt{\frac{9}{M}\log \frac{M}{\delta}}\left(\sqrt{\gamma t^{1-\theta}} \vee 1\right)\right) + \sqrt{B}\right) \times$$

$$\times \left(\frac{8}{(1-\theta)} + 4\log t + 4 + \sqrt{2}\gamma\right)\left(\frac{\sqrt{\gamma t^{1-\theta}}}{n} + \frac{\sqrt{2\sqrt{p}q_0\mathcal{N}(\frac{\kappa^2}{\gamma t^{1-\theta}})}}{\sqrt{n}}\right)\log \frac{4}{\delta}, \quad (66)$$

*where $q_0 = \max\left(2.55, \frac{2\kappa^2}{\|L\|}\right)$.*

*Proof.* The proof can be derived from the one of Theorem 5 in [33] with $\lambda = \frac{1}{\gamma_t t}$, $\delta_1 = \delta_2 = \delta$, and some changes we now describe. Instead of the iteration $\nu_t$ and $\mu_t$ defined in [33] we consider (40) and (41) respectively, as well as the operators $S_M, C_M, L_M, \widehat{S}_M, \widehat{C}_M, \widehat{L}_M$ defined in Section 2 instead of $S_\rho, \mathcal{T}_\rho, \mathcal{L}_\rho, S_\mathbf{x}, \mathcal{T}_\mathbf{x}, \mathcal{L}_\mathbf{x}$ defined in [33]. Instead of assuming that exists a $\kappa \geq 1$ for which $\langle x, x' \rangle \leq \kappa^2, \forall x, x' \in X$ we have Assumption 2 which imply the same $\|C_M\| \leq \kappa^2$ upper bound of the operators used in the proof. Further, when in the proof we need to bound $\|v_{t+1}\|$ we use our Lemma 2 instead of Lemma 16 of [33]. In addition instead of Lemma 18 of [33] we use Lemma 6 of [8], together with Lemma 4, obtaining the desired result with probability $1 - 3\delta$, when $M$ satisfies $M \geq (4 + 18\gamma_t t) \log \frac{12\gamma_t t}{\delta}$. Under the assumption that $\gamma_t = \gamma\kappa^{-2}t^{-\theta}$, the two condition above can be rewritten as (65). $\qquad\square$

The next lemma states that the third term (50) of the error decomposition is equal to zero.

**Lemma 7.** *Under Assumption 3 the following holds for any $t, M, n \in \mathbb{N}$*

$$\|S_M\widetilde{v}_t - S_Mv_t\| = 0 \quad a.s. \tag{67}$$

*Proof.* From Lemma 1 and the definition of operator norm the result follows trivially. $\qquad\square$

The next Lemma is a known result from Lemma 8 of [8] which bounds the distance between the Tikhonov solution with RF and the Tikhonov solution without RF (52).

**Lemma 8.** *Under Assumption 2 and 3 for any $\lambda > 0$, $\delta \in (0, 1/2]$, when*

$$M \geq \left(4 + \frac{18\kappa^2}{\lambda}\right) \log \frac{8\kappa^2}{\lambda\delta} \tag{68}$$

*the following holds with probability at least $1 - 2\delta$,*

$$\|LL_\lambda^{-1}Pf_\rho - L_ML_{M,\lambda}^{-1}Pf_\rho\| \leq 4R\kappa^{2r}\left(\frac{\log\frac{2}{\delta}}{M^r} + \sqrt{\frac{\lambda^{2r-1}\mathcal{N}(\lambda)^{2r-1}\log\frac{2}{\delta}}{M}}\right)q^{1-r}, \tag{69}$$

*where $q := \log \frac{11\kappa^2}{\lambda}$.*

The next lemma is one of our main contributions and studies how the population gradient decent with RF and ridge regression with RF are related (51).

**Lemma 9.** *Under Assumption 3 the following holds with probability $1 - \delta$ for $\lambda = \frac{1}{\sum_{i=1}^t \gamma_i}$ for all $t \in [T]$*

$$\|S_Mv_{t+1} - L_ML_{M,\lambda}^{-1}Pf_\rho\|_\rho \leq 8R\kappa^{2r}\left(\frac{\log\frac{2}{\delta}}{M^r} + \sqrt{\frac{\mathcal{N}((\sum_{i=1}^t\gamma_i)^{-1})^{2r-1}\log\frac{2}{\delta}}{M(\sum_{i=1}^t\gamma_i)^{2r-1}}}\right) \times$$

$$\times \log^{1-r}\left(11\kappa^2\sum_{i=1}^t\gamma_i\right) + 2R\left(\sum_{i=1}^t\gamma_i\right)^{-r}, \tag{70}$$

*when*

$$M \geq \left(4 + 18\sum_{i=1}^t\gamma_i\right)\log\left(\frac{8\kappa^2\sum_{i=1}^t\gamma_i}{\delta}\right). \tag{71}$$

*Proof.* Denoting $Q_M = \sum_{i=1}^t \gamma_i \prod_{k=i+1}^t (I - \gamma_kL_M)$ we can write

$$S_Mv_{t+1} = Q_ML_MPf_\rho$$

Then

$$S_Mv_{t+1} - L_ML_{M,\lambda}^{-1}Pf_\rho = Q_ML_{M,\lambda}L_ML_{M,\lambda}^{-1} - L_ML_{M,\lambda}^{-1}Pf_\rho$$

$$= (Q_M(L_M + \lambda I) - I)L_ML_{M,\lambda}^{-1}Pf_\rho. \tag{72}$$

Denote by $A_{i,t}$ the operator $A_{i,t} := \prod_{k=i}^{t}(I - \gamma_k L_M)$, and note that
$$A_{i,t} := (I - \gamma_k L_M)A_{i+1,t}.$$

We can then derive
$$Q_M L_M = \sum_{i=1}^{t}\gamma_i \prod_{k=i+1}^{t}(I - \gamma_k L_M)L_M = \sum_{i=1}^{t}(I - (I - \gamma_i L_M)) \prod_{k=i+1}^{t}(I - \gamma_k L_M)$$
$$= \sum_{i=1}^{t}(I - (I - \gamma_i L_M))A_{i+1,t} = \sum_{i=1}^{t}A_{i+1,t} - \sum_{i=1}^{t}(I - \gamma_i L_M)A_{i+1,t}$$
$$= \sum_{i=1}^{t}A_{i+1,t} - \sum_{i=1}^{t}A_{i,t} = I + \sum_{i=2}^{t}A_{i,t} - \sum_{i=1}^{t}A_{i,t} = I - A_{1,t}.$$

We now write
$$\|(Q_M(L_M + \lambda I) - I)L_M\| = \|(Q_M L_M + \lambda Q_M - I)L_M\|$$
$$= \|(I - A_{1,t} + \lambda Q_M - I)L_M\|$$
$$= \|\lambda Q_M L_M - A_{1,t}L_M\|$$
$$\leq \|\lambda Q_M L_M\| + \|A_{1,t}L_M\|. \tag{73}$$

For the first term in (73) we have
$$\|\lambda Q_M L_M\| = \lambda\|I - A_{1,t}\| \leq \lambda,$$

since $L_M$ is positive operator and $\gamma_i\|L_M\| < 1$, so $A_{1,t}$ is positive with norm smaller than one by construction, implying that $\|I - A_{1,t}\| \leq 1$. The second term in (73) can be bounded using Lemma 15 in [33],
$$\|A_{1,t}L_M\| \leq (\sum_{i=1}^{t}\gamma_i)^{-1}$$

Now back to (72), we can write
$$\|S_M v_{t+1} - L_M L_{M,\lambda}^{-1}Pf_\rho\|_\rho \leq \left(\lambda + \frac{1}{\sum_{i=1}^{t}\gamma_i}\right)\|L_{M\lambda}^{-1}Pf_\rho\|_\rho. \tag{74}$$

Setting $\lambda = \frac{1}{\sum_{i=1}^{t}\gamma_i}$, and calling this quantity $\widetilde{\lambda}$ for the rest of the proof, we can write
$$\|S_M v_{t+1} - L_M L_{M,\widetilde{\lambda}}^{-1}Pf_\rho\|_\rho \leq 2\widetilde{\lambda}\|L_{M\widetilde{\lambda}}^{-1}Pf_\rho\|_\rho \tag{75}$$
$$= 2\|(\widetilde{\lambda}L_{M\widetilde{\lambda}}^{-1} - \widetilde{\lambda}L_{\widetilde{\lambda}}^{-1} + \widetilde{\lambda}L_{\widetilde{\lambda}}^{-1})Pf_\rho\|_\rho \tag{76}$$
$$\leq 2\|(\widetilde{\lambda}L_{M\widetilde{\lambda}}^{-1} - \widetilde{\lambda}L_{\widetilde{\lambda}}^{-1})Pf_\rho\|_\rho + 2\widetilde{\lambda}\|L_{\widetilde{\lambda}}^{-1}Pf_\rho\|_\rho. \tag{77}$$

Since $AA_\lambda^{-1} = I - \lambda A_\lambda^{-1}$ for any bounded symmetric operator $A$ and $\lambda > 0$, we can write the last term of (77) as
$$\widetilde{\lambda}\|L_{\widetilde{\lambda}}^{-1}Pf_\rho\|_\rho = \|(LL_{\widetilde{\lambda}}^{-1} - I)Pf_\rho\|_\rho.$$

We can then use Lemma 10 to control this quantity as
$$\|(LL_{\widetilde{\lambda}}^{-1} - I)Pf_\rho\|_\rho \leq R\widetilde{\lambda}^r. \tag{78}$$

For the first term, analogously
$$\|(\widetilde{\lambda}L_{M\widetilde{\lambda}}^{-1} - \widetilde{\lambda}L_{\widetilde{\lambda}}^{-1})Pf_\rho\|_\rho = \|((I - \widetilde{\lambda}L_{M\widetilde{\lambda}}^{-1}) - (I - \widetilde{\lambda}L_{\widetilde{\lambda}}^{-1}))Pf_\rho\|_\rho$$
$$= \|(L_M L_{M\widetilde{\lambda}}^{-1} - LL_{\widetilde{\lambda}}^{-1})Pf_\rho\|_\rho$$
$$\leq 4R\kappa^{2r}\left(\frac{\log\frac{2}{\delta}}{M^r} + \sqrt{\frac{\widetilde{\lambda}^{2r-1}\mathcal{N}(\widetilde{\lambda})^{2r-1}\log\frac{2}{\delta}}{M}}\right)\left(\log\frac{11\kappa^2}{\widetilde{\lambda}}\right)^{1-r}, \tag{79}$$

where the last step holds when $M \geq (4 + 18\widetilde{\lambda}^{-1})\log(8\kappa^2(\widetilde{\lambda}\delta)^{-1})$ and consists in the application of Lemma 9. Now recalling the definition of $\widetilde{\lambda}$ we complete the proof. $\qquad\square$

The last result is a classical bound of the approximation error for the Tikhonov filter (53), see [11].

**Lemma 10** (From [11] or Lemma 5 of [8]). *Under Assumption 3*

$$\|LL_\lambda^{-1}Pf_\rho - Pf_\rho\| \le R\lambda^r \tag{80}$$

## A.4   Proofs of Theorems

We now present the proofs of our theorems. Theorem 2 and 1 are specific case of the more general Theorem 3.

**Proof of Theorem 3.** We start considering Lemma 6, and we note that condition (65) is satisfied when

$$M \ge \left(4 + 18\gamma T^{1-\theta}\right)\log\frac{12\gamma T^{1-\theta}}{\delta}. \tag{81}$$

Noting that (19) imply $\sqrt{2}\gamma \le 1$, we can derive from (66)

$$\|S_M(\widehat{v}_{t+1} - v_{t+1})\|^2 \le \left(\frac{(17 - 9\theta)\sqrt{8\sqrt{p}}}{(1 - \theta)}\right)^2 \times$$
$$\times \left(32B + 64R^2\kappa^{4r}\left(1 + \frac{9}{M}\log\frac{M}{\delta}\left(\gamma t^{1-\theta}\vee 1\right)\right)\right) \times$$
$$\times \frac{q_0\mathcal{N}(\frac{\kappa^2}{\gamma t^{1-\theta}})}{n}\left(\log^2 t \vee 1\right)\log^2\frac{4}{\delta}, \tag{82}$$

when (81) holds.

Let $\lambda = \frac{\kappa^2}{\gamma t^{1-\theta}}$. Given Lemma 8 we derive from (69) that

$$\left\|LL_\lambda^{-1}Pf_\rho - L_M L_{M,\lambda}^{-1}Pf_\rho\right\|^2 \le 32R^2\kappa^{4r}\left(\frac{\log^2\frac{2}{\delta}}{M^{2r}} + \frac{\mathcal{N}(\frac{\kappa^2}{\gamma t^{1-\theta}})^{2r-1}\log\frac{2}{\delta}}{M(\gamma t^{1-\theta}\kappa^{-2})^{2r-1}}\right) \times$$
$$\times \log^{2-2r}\left(11\gamma t^{1-\theta}\right), \tag{83}$$

when (81) holds.

Let $\gamma_t = \gamma\kappa^{-2}t^{-\theta}$ for all $t \in [T]$. Given Lemma 9 we derive from (70)

$$\left\|S_M v_{t+1} - L_M L_{M,\lambda}^{-1}Pf_\rho\right\|^2 \le 8R^2\kappa^{4r}\left(32\left(\frac{\log^2\frac{2}{\delta}}{M^{2r}} + \frac{\mathcal{N}(\frac{\kappa^2}{\gamma t^{1-\theta}})^{2r-1}\log\frac{2}{\delta}}{M(\gamma t^{1-\theta}\kappa^{-2})^{2r-1}}\right) \times \right.$$
$$\left. \times \log^{2-2r}\left(11\gamma t^{1-\theta}\right) + \left(\frac{1}{\gamma t^{1-\theta}}\right)^{2r}\right), \tag{84}$$

when (81) holds.

Similarly from Lemma 10

$$\|LL_\lambda^{-1}Pf_\rho - Pf_\rho\|^2 \le R^2\kappa^{4r}\left(\frac{1}{\gamma t^{1-\theta}}\right)^{2r}. \tag{85}$$

The desired result is obtained by gathering the results in (64), (82), (84), (83), (85). Requiring $\gamma, M$ to satisfy the associated conditions (81), (62), (63). In particular note that (62) is satisfied when $\theta = 0$ by $\gamma \le (8(\log T + 1))^{-1}$, while, if $\theta > 0$, we have

$$\frac{t^{\min(\theta,1-\theta)}}{8(\log t + 1)} = e^{-\min(\theta,1-\theta)}\frac{(et)^{\min(\theta,1-\theta)}}{8\log(et)} \ge e^{-\min(\theta,1-\theta)}\inf_{t\in 1}\frac{(et)^{\min(\theta,1-\theta)}}{8\log(et)}$$
$$= e^{-\min(\theta,1-\theta)}\inf_{z\ge e^{\min(\theta,1-\theta)}}\frac{z}{\frac{8}{\min(\theta,1-\theta)}\log z}$$
$$\ge e^{-\min(\theta,1-\theta)}\inf_{z\ge 1}\frac{z}{\frac{8}{\min(\theta,1-\theta)}\log z} \ge e^{-\min(\theta,1-\theta)}\frac{\min(\theta,1-\theta)}{4},$$

where we performed the change of variable $t^{\min(\theta,1-\theta)} = z$. Finally note that $e^{-\min(\theta,1-\theta)} \geq e^{-1/2}$, for any $\theta \in (0,1)$. Moreover the (81), (63) are satisfied for any $t \in [T]$ by requiring them to hold for $t = T$. $\qquad\square$

**Proof of Theorem 2.** Choosing $\theta = 0$ in Theorem 3 we complete the proof. $\qquad\square$

**Proof of Theorem 1.** Considering the case of Assumption 5 with $\alpha = 1$ we can bound $\mathcal{N}(1/\gamma t) \leq \gamma t$ in Theorem 3 and complete the proof. $\qquad\square$