[Reviews · NeurIPS 2018]

Reviewer 1



After Author Response: Thanks to the authors for their various clarifications. I have updated my score to an 8 accordingly. I think the planned updates to the empirical section will add a lot of value. ============= Summary: This paper analyzes the generalization performance of models trained using mini-batch stochastic gradient methods with random features (eg, for kernel approximation), for regression tasks using the least squares loss. Their main theorem (Theorem 1) bounds the gap in generalization performance between the lowest risk model in the RKHS with the SGD trained model after t mini-batch updates; the bound is in terms of the learning rate, the mini-batch size, the number of random features M, the training set size n, and t. They show that under certain choices of these parameters, M = O(sqrt(n)) features are sufficient to guarantee that the gap is <= 1/sqrt(n). This is the learning rate achieved by the exact kernel model, which has been proven to be optimal. This shows that instead of paying O(n^3) time and O(n^2) space to solve the exact kernel problem, one can spend O(dn * sqrt(n)) time and O(sqrt(n)d) space and solve the problem using random features and stochastic gradient descent. These are significant savings. The theorem also suggests that the learning rate should increase as the mini-batch size increases; this is validated empirically. The paper also presents experiments showing that classification error saturates for M << n. The work most relevant to this paper is the paper from NIPS of last year by Rudi and Rosasco: "Generalization Properties of Learning with Random Features" (I will refer to this paper as [RR17]). That paper proves a result quite similar to the one in this paper, but specifically for the kernel ridge regression estimator learned with random features (closed form solution, Equation 7 in that paper). The present work extends that result to the model learned using mini-batch SGD with random features. This is significant because in the large scale setting SGD is typically the method of choice. Solving for the KRR estimator requires storing the full random features covariance matrix in memory in order to invert it; this could be prohibitive when the number of random features is very large. Stochastic methods are thus better suited for the large-scale setting. I will now discuss the quality, clarity, originality, and significance of this work: 1) Quality: Quality of the work is high. The paper solves a challenging theoretical question using sophisticated analysis. The experiments are a weaker point of the paper. Figure 1b is ok, showing how larger mini-batches should be paired with larger learning rates; though perhaps each column should be converted into a line on a plot. Figure 1a in my opinion could be presented a lot more clearly; I would prefer seeing plots on various datasets showing how performance varies as you increase the number of random features (after training has finished, for example). Showing results using color-scale for all 10 passes of the data is not a good use of space in my opinion. 2) Clarity: The work is presented quite clearly. For example, I appreciated how the corollaries explained how the various parameters could be set to achieve the desired rates. I also liked how in lines 155-162 describes clearly what happens as you increase the batch size/learning rate. 3) Originality: This work is relatively original. It extends recent results [RR17] to a new training setting. I'm not certain how novel the proof techniques are. 4) Significance: I believe this work is an important step toward understanding the performance of models learned using random features with mini-batch SGD in general (the results in this paper only apply to least squares loss). It appears that in practice learning models using SGD and random features is very effective; this work helps explain why. This work also provides a way of understand how different SGD hyperparameters should be chosen together in order to achieve small excess risk --- this insight is relevant in practice. Questions: -- Couldn't a result similar to the one in this paper follow pretty directly from the [RR17] result? In particular, SGD on the regularized least squares objective should converge to the optimum at a certain rate; the optimum of this objective is precisely given by the closed form ridge regression solution, for which the [RR17] results presents generalization bounds. -- Could these results be extended to fastfood features? That would reduce the time to O(log(d) n * sqrt(n)) and the space to O(sqrt(n)). NIT: - Section 2.1: 'D' is used to denote dimension of data space X. In section 2 (lines 68-73) 'd' is used. - Section 3.2: 'L' is never defined. And in the equation for Cf(x), there should be an "f" before the "(x')\rho_X(x')". - Lines 188-189: "The smaller is \alpha the more stringent...". I believe it should be: "The larger \alpha is, the more stringent..." - I don't think the acronym SGM is ever defined (stochastic gradient methods?) - Line 244: SMG should be SGM. - Lines 35-36 should cite [RR17] [RR17] Alessandro Rudi, Lorenzo Rosasco: Generalization Properties of Learning with Random Features. NIPS 2017

Reviewer 2



Edit: I read the author response. The additions and clarifications they promise to make further strengthen my opinion that this is a very good submission that will be of value to the community. The work presents an analysis of the impact of the hyperparameters in stochastic gradient (SG) optimization on the excess risk of empirical risk minimization problems using random features to approximately minimize least squares regression problems in a Hilbert space. This is part of a recent stream of work at NIPS level conferences that has investigated the generalization error of least squares problems in Hilbert spaces using dimensionality reduction techniques (Nystrom, uniform sampling, random features, etc.). To my knowledge the work is novel in that it characterizes the dependence of the excess risk on specific hyperparameters of SG -- iterations, step-size, mini-batch size--- in addition to the number of random features used. Prior work that I have seen focused more on the impact of the number of random features than the impact of SG using the random features. All the works I would have expected to seen referenced are there. Strengths: - a clean characterization of the excess risk of SG using random features in terms of specific hyperparameters of the algorithm - the proof sketch is straightforward and gives the reader insight into how the effects of the different hyperparameters were teased out - the paper is well-written in that it gets across the results and the ideas of the analysis in a way that should be understandable to a NIPS audience (as opposed to say COLT), and the authors clearly state in Corollaries 1 and 2 what concrete hyperparameter choices are suggested by their theory - a short experimental section shows that the theory bears out in practice Weakness: - would like to have seen a discussion of how these results related to the lower bounds on kernel learning using low-rank approximation given in "On the Complexity of Learning with Kernels". - In Assumption 5, the operator L is undefined. Should that be C?

Reviewer 3



Edit: My score remains the same because no additional experimental results are shown at the time of author rebuttal. Summary This paper proved a novel error bound of the solution of kernel ridge regression learned by stochastic gradient descent and random features approximation. The first theorem proved the high probability bound calculated from batch size, learning rate, iterations, sample size and the dimension of approximation feature. They demonstrated the several settings that only O(N^(1/2)) features are required to ensure O(N^(-1/2)) error, resulting in O(N^(3/2)D) time complexity and O(N^(1/2)D) space complexity. This is smaller than existing O(N^2) time complexity and O(N^(3/2)) space complexity when D<